# COVID-19 Vaccine Hesitancy among Italian Healthcare Workers: Latent Profiles and Their Relationships to Predictors and Outcome

**DOI:** 10.3390/vaccines11020273

**Published:** 2023-01-27

**Authors:** Igor Portoghese, Melinda Siddi, Luchino Chessa, Giulia Costanzo, Vanessa Garcia-Larsen, Andrea Perra, Roberto Littera, Giada Sambugaro, Stefano Del Giacco, Marcello Campagna, Davide Firinu

**Affiliations:** 1Department of Medical Sciences and Public Health, University of Cagliari, 09100 Cagliari, Italy; 2Association for Advancing on Transplantation Research. O.d.V., Non Profit Organisation, 09100 Cagliari, Italy; 3Department of International Health, Bloomberg School of Public Health, The Johns Hopkins University, Baltimore, MD 21205, USA; 4Department of Biomedical Sciences, University of Cagliari, 09100 Cagliari, Italy; 5Medical Genetics, Department of Medical Sciences and Public Health, University of Cagliari, 09100 Cagliari, Italy

**Keywords:** vaccine hesitancy, latent profiles, healthcare workers, conspiracy beliefs, COVID-19

## Abstract

Vaccine hesitancy and conspiracy beliefs among healthcare workers (HCWs) represent operational priorities that require urgent attention. Identifying and classifying specific subpopulation of hesitancy is crucial to customize educational and intervention strategies to enhance the acceptance and uptake rate of vaccination. Thus, the main purpose of our study was to empirically identify latent profiles of vaccine hesitancy among Italian HCWs adopting a person-centered approach and investigating their relationships with antecedents and intention to get a fourth dose of COVID-19 vaccine. We conducted latent profile analyses (LPA) to identify different configurations of vaccine hesitancy based on five antecedents of vaccination: confidence, complacency, constraints, calculation, and collective responsibility among a sample of Italian HCWs (*n* = 573). LPA revealed four distinct profiles: believer (61.5%), middler (24.7%), hesitant (9.00%), and rejecter (4.7%). Having conspiracy beliefs was associated with a greater likelihood of membership in all but believer. Finally, the likelihood of intention to get a fourth dose of COVID-19 vaccine was lowest in the rejector and hesitant profiles. Theoretical contributions and implications for practice are discussed.

## 1. Introduction

Vaccination is one of the cheapest and most successful interventions to improve and protect the health of the general population. According to the World Health Organization (WHO) [1], vaccination saves 2 to 3 million lives worldwide every year. Concerning the coronavirus disease 2019 (COVID-19), the efficacy of vaccines in preventing severe outcomes such as hospitalization and death has been documented in a recent literature review and meta-analysis [2]. Particularly, immunization of healthcare workers (HCWs) is considered a crucial infection control measure with the potential to protect both HCWs and patients [3,4]. However, a major hindrance to the impact of vaccination is the low uptake of vaccines [5]. Analyzing COVID-19 vaccination acceptance rates, what emerges is that low rates were reported in the Middle East, Russia, Africa, and several European countries [6]. In a recent metanalysis, Wang and colleagues found that the global pooled acceptance rate of COVID-19 vaccination was 67.8% (95% CI: 67.1–68.6) among the general population, and 67.5% (95% CI: 64.4–70.6) among HCWs [7]. For example, recently, a systematic review reported that vaccine acceptance amongst HCWs correlated with their willingness to recommend the COVID-19 vaccination to their patients [8]. In this sense, it is necessary to investigate HCWs’ COVID-19 vaccine hesitancy to better understand the barriers, especially considering the last WHO’s global COVID-19 vaccination strategy [9]. In fact, among the strategies, the WHO considered continued investments in vaccination coverage, including boosters, as operational priorities that require urgent attention [9]. Receiving a booster dose of COVID-19 vaccines has been proved to significantly increase immunogenicity and to improve the peak antibody levels following the primary immunization series among healthy adults [10,11].

In the last 10 years, many studies have focused on vaccine hesitancy and its global impact on public health, defining vaccine hesitancy in different ways [12]. The WHO’s Strategic Advisory Group of Experts (SAGE) Working Group described vaccine hesitancy as “a complex behavioural phenomenon specific to vaccines, context, time, and place and influenced by factors of complacency, convenience, and confidence” [13] (p. 4162). Accordingly, complacency, convenience, and confidence are the three main antecedents of vaccine hesitancy as postulated in the 3C model proposed by the WHO EURO Vaccine Communications Working Group [13]. In the 3C model, three main antecedents of vaccine hesitancy are identified: (1) complacency, linked to not perceiving diseases as high risk and vaccination as necessary, (2) convenience, referred to as the practical barriers to receive the vaccination, and (3) confidence concerns, understood as a lack of trust in safety and effectiveness of vaccines [13]. Recently, Betsch and colleagues proposed an extended version of the 3C model, developing the 5C model of psychological antecedents of vaccination, adding calculation and collective responsibility [14]. Specifically, calculation “refers to individuals’ engagement in extensive information searching” and is “related to perceived vaccination and disease risks” [14] (p. 6). Collective responsibility refers to the “willingness to protect others by one’s own vaccination by means of herd immunity” [14] (p. 7).

Moreover, conspiracy beliefs represent another important barrier that may be associated with negative attitudes toward vaccination, such as vaccine hesitancy. Accordingly, conspiracy theories “explain the ultimate causes of significant events as the secret actions of malevolent groups, who cover up information to suit their own interests” [15] (p. 459). The dominant vaccination conspiracy theories claim that vaccination data are systematically falsified, the adverse side effects of vaccines are deliberately hidden from the public, and that this is done to ensure profits for pharmaceutical companies and governments [15]. Recently, two systematic reviews found that COVID-19 conspiracy beliefs were a significant predictor of COVID-19 vaccination hesitancy among the general population [16,17]. Among HCWs, Al-Sanafi and Sallam found that conspiracy beliefs were significantly associated with less intention to receive COVID-19 vaccination [18].

One of the main purposes of the vaccine hesitancy literature is to identify and classify people for targeting specific interventions [19,20,21]. In fact, evidence suggests that it is crucial to customize intervention strategies targeting a specific subpopulation to enhance the acceptance and uptake rate of vaccination [22,23,24]. In this sense, the empirical identification of subpopulations and their characteristics with regards to vaccine hesitancy will serve as key components of a successful vaccination strategy. Very few studies have employed a person-centered approach in classifying people into subpopulations depending on their vaccination hesitancy [23,25]. A recent review on COVID-19 vaccination hesitancy among HCWs, showed that different attitude patterns and behaviors against COVID-19 vaccination could be identified [26]. These patterns may combine and define different subpopulations of hesitant individuals. For example, in their recent study, Zhang and colleagues adopted the 5C model and identified four latent profiles of hesitant among nurses: believers (high confidence and collective responsibility), free riders (similar characteristics to believers, except for a low collective responsibility), middlemen (average levels in all 5C factors), and contradictors (high in all 5C factors) [25].

In another similar study performed using the same 5C framework in the same working population (nurses), Leung and colleagues identified five latent profiles of hesitant: believers (high confidence, collective responsibility; low complacency, constraint), skeptics (opposite to the believers), outsiders (low calculation, collective responsibility), contradictors (high in all 5C constructs), and middlers (average levels in all 5C factors) [23]. In line with these results, Kwok and colleagues identified four latent profiles: believers (high confidence, high collective responsibility, low complacency, and low constraint), fence-sitters (average scores in all 5C factors), apathetics (low in calculation, high in complacency, and high constraint), and skeptics (low confidence, low collective responsibility, high complacency, and high constraint) [27].

Thus, the main aim of this study was to empirically identify latent profiles of the 5C psychological antecedents of vaccine hesitancy (confidence, complacency, constraints, calculation, and collective responsibility) among Italian HCWs. A latent profile analysis (LPA) will be used, investigating profiles’ relationships with antecedents and intention to get a fourth dose of COVID-19 vaccine.

## 2. Materials and Methods

Adopting a convenience-snowball sampling method, a self-reported online survey was implemented in Italy from 28 April to 30 June 2022, using Limesurvey and promoted by sharing the link on social network platforms (Facebook, LinkedIn, and Twitter). The only inclusion criterion for this convenience sample was to be a healthcare worker in Italy. The survey’s homepage illustrated the online informed consent form with detailed information about the study purpose, the questionnaire general description, including information about risks and benefits of participating in the survey. Anonymity was assured because no IP address was registered, and no sensitive data were requested. The survey was organized into two sections: (1) demographics data including gender, age, geographic area of employment, marital status, profession, years of practice, vaccines history (Flu and Hepatitis B [HBV]), serological tests (in the last 24 months), having family\friends\colleagues have died of COVID-19 (yes\no), and intention to get a fourth dose of COVID-19 vaccine (yes\no\I do not know); and (2) psychological antecedents of vaccine hesitancy and Vaccine Conspiracy Belief Scale (VCBS) [28]. A total of 573 healthcare workers participated into the study.

### 2.1. Measures

We translated the English-based measures into Italian following the translation/back-translation procedure to guarantee semantic equivalence [29]: Firstly, items were translated to Italian by a panel of three experts. This was followed by peer back-translation to English version by both first and second authors, comparing the original English version and back-translated English version questionnaires to examine any discrepancies between the two versions.

*Psychological antecedents of vaccine hesitancy* [14]. The 5C psychological antecedents to vaccination scale is composed of 15 items and covers five antecedents of vaccination: confidence (three items, ω = 0.905), complacency (three items, ω = 0.789), constraints (three items, ω = 0.715), calculation (three items, ω = 0.779), and collective responsibility (three items, ω = 0.879). Each item was rated on a seven-point Likert scale from 1 = strongly disagree to 7 = strongly agree.

*Vaccine Conspiracy Belief Scale* (VCBS) [27]. The VCBS was used to measure conspiracy belief and includes seven items (ω = 0.971). Each item was rated on a seven-point Likert scale from 1 = strongly disagree to 7 = strongly agree.

### 2.2. Analyses

#### 2.2.1. Measurement Models

Preliminary measurement models (confirmatory factor analysis; CFA) were used to assess the multidimensional factor structure of 5C measure using the robust maximum likelihood (MLR) in Mplus 8.8 [30]. Specifically, we contrasted three models: (1) model in which the measures of all 5C factors were set to load on their respective factors, (2) a more constrained model in which most correlated factors were set to load on a combined factor, and (3) a model where all items were set to load on a single general factor. The overall model fit was assessed using the following fit indices (cut-off points): the Comparative Fit Index (CFI ≥ 0.90), the Tucker–Lewis Index (TLI ≥ 0.90), the root mean square error of approximation (RMSEA ≤ 0.06), and the standardized root mean square residual (SRMR ≤ 0.06) [31]. For measures’ reliability, we assessed omega (ω) coefficient.

#### 2.2.2. Latent Profile Analyses

We used the factor scores derived from the best fitting measurement model (CFA) for conducting the latent profile enumeration [32]. LPA was conducted using 1000 random starts, 250 final stage optimizations, and 50 initial stage iterations. The final model was estimated with 6000 random starting values, 1000 iterations, and 200 final stage optimizations to ensure a local maximum was not found [33,34]. The final number of latent profiles were selected based on multiple statistical indices, theoretical interpretability, and substantive meaningfulness [35,36].

Statistical indices included minimum values of Akaike Information Criterion (AIC), Bayesian Information Criterion (BIC), and sample-size adjusted BIC (aBIC), with smaller values indicating more parsimony when comparing models [37]. Additionally, because these indices often continue to improve with the addition of latent profiles, particularly with large sample sizes [35], and we also examined elbow plots, or graphical representations of the AIC and BIC values [38]. We also used the Vuong–Lo–Mendell–Rubin likelihood ratio test (LRT) to compare the k latent class model against the k-1 latent class model, whereby the k-1 latent class model would be preferred when the test shows a non-significant p-value [39]. Finally, we examined the entropy values [40], which provides information about classification probabilities, with values closer to 1 indicating greater classification reliability and precision.

#### 2.2.3. Predictors and Outcome of Profiles

Vaccines history (Flu and HBV), serological tests (in the last 24 months), having family\friends\colleagues have died of COVID-19, and Vaccine Conspiracy Belief were assessed as predictors of profile membership using the R3STEP function in Mplus [41,42]. This estimates a multinomial logistic regression (log odds that represent the k-1 regression coefficient in relation to a reference profile) with the categorical latent profile as the outcome to examine how each predictor, accounting for the others, related to the likelihood of membership in the hesitancy profiles. The odds ratios represent the transformed log odds that indicate a students’ likelihood of profile membership. Finally, intention to get a fourth dose of COVID-19 vaccine (even if it will be not mandatory) and were assessed between profiles using the DCAT function in Mplus, and the Wald chi-square test was used to estimate the equality of means between profiles [41].

### 2.3. Ethical Aspects

The study was an extension of the CORIMUN study protocol at the Teaching Hospital of the Cagliari University. The study, including informed consent procedures, conforms to the ethical guidelines of the Declaration of Helsinki and was approved by the Ethics Committee of the Cagliari University Hospital on 27 May 2020 (protocol GT/2020/10894) and its extension of 27 January 2021.

## 3. Results

### 3.1. Preliminary Analyses

As a first step, we tested the five-factor measurement model for the 5C of psychological antecedents of vaccine hesitancy. We contrasted the five-factors model with two alternative models. The six-factor CFA did provide a good degree of fit to the data according to the CFI = 0.959, TLI = 0.951, the RMSEA = 0.047, and the SRMR = 0.040 (see Appendix A).

### 3.2. Latent Profile Analysis

Fit indices resulting from the latent profile models containing up to eight profiles are provided in Table 1. BIC decreased as the number of profiles increased, but the decrease became minimal beginning from the four-profile to the eight-profile model. However, information criteria did not reach a minimum value, therefore we relied on the elbow plot (Figure 1).

The four-profiles solution of 5C are illustrated in Figure 2. The first profile contained 61.6% (*n* = 351; latent profile membership probability = 0.98) of the participants and was characterized by slightly below average levels of Calculation, highest levels of Collective Responsibility and Confidence, and lowest levels of Complacency and Constraints. We labeled this profile “believer”. The second profile contained 24.7% (*n* = 141; latent profile membership probability = 0.94) of the participants and was characterized by about-average levels of Calculation and Collective Responsibility, above-average levels of Complacency, low levels of Confidence, and above-average levels of and Constraints. We labeled this profile “middler”. The third profile contained 9.0% (*n* = 51; latent profile membership probability = 0.97) of the participants and was characterized by above-average levels of Calculation, low levels of Collective Responsibility and Confidence, and high levels of Complacency and Constraints. We labeled this profile “hesitant”. Finally, the fourth profile contained 4.7% (*n* = 27; latent profile membership probability = 0.99) of the participants and was characterized by highest levels of Calculation, Complacency and Constraints and lowest levels of Collective Responsibility and Confidence. We labeled this profile “rejector”.

### 3.3. Predictors and Profile Outcomes

We then applied the R3STEP command by including auxiliary variables for testing the relation between vaccines history (Flu and HBV), serological testing for SARS-CoV-2 specific antibody (in the last 24 months), having had family\friends\colleagues died of COVID-19, conspiracy beliefs, and the four identified profiles. Table 2 shows the results of the multinomial logistic regression models predicting profile membership. Specifically, having an history of Flu vaccination was associated with a lesser likelihood of membership in hesitant profile than the believer profile (B = −1.21, s.e. = 0.61, *p* < 0.05, OR = 3.36). Furthermore, having an history of HBV vaccination was associated with a greater likelihood of membership in believer profile (B = 2.49, s.e. = 1.01, *p* < 0.05, OR = 12.05) and middler (B = 2.00, s.e. = 0.93, *p* < 0.01, OR = 7.41) profiles than the rejectr profile.

Concerning serological tests (in the last 24 months), have had a test in the last 24 months was associated with a greater likelihood of membership in middler profile than the believer profile (B = 0.80, s.e. = 0.37, *p* < 0.05, OR = 2.23).

Concerning having family\friends\colleagues that have died of COVID-19, it was associated with a greater likelihood of membership in middler profile (B = 1.62, s.e. = 0.87, *p* < 0.05, OR = 5.07) and hesitant (B = 1.80, s.e. = 0.72, *p* < 0.05, OR = 6.08) profiles than the rejectors profile. Finally, having conspiracy beliefs was associated with a greater likelihood of membership in all but believer profile.

Concerning the outcome variable (Table 3 and Table 4), the likelihood of intention to get a fourth dose of COVID-19 vaccine (even if not mandatory), was highest for believer profile (80.4%, s.e. = 0.03), followed by middler profile (48.3%, s.e. = 0.04). The lowest likelihood to not get the vaccine was in the rejectors profile (100%, s.e. = 0.00), followed by the hesitant profile (67.8%, s.e. = 0.07).

## 4. Discussion

To our knowledge, this is the first study to investigate latent profiles of hesitancy towards booster dose of COVID-19 vaccine among Italian HCWs. Taking a person-centered approach, we empirically identified four different profiles of HCWs based on their scores in 5Cs measure [14]. The majority of our population of HCWs falls into the believer profile, converging with results from previous research, defined by the highest levels of confidence and collective responsibility, and lowest levels in complacency and constraints [23,25,27]. Furthermore, in line with Zhang and colleagues [25], in our study, calculation was slightly below average levels, and it was the profile with the lowest levels of calculation. These results contradict the 5C model assumptions about the role of the calculation in the vaccine hesitancy [14]. In fact, according to the model, engaging in high calculation is considered a risk-averse attitude correlated to higher intention to vaccinate. However, despite this assumption, our results suggest that high calculation is not an exclusive characteristic of believers. The second profile was the middler, characterized by about average levels of all 5C components and was partially in line with the previous literature [23,25]. More specifically, we found that this profile showed a configuration of the 5Cs that are not extreme in their values, with true means above 5 (agree point on the response scale) for calculation and confidence, and 6 for collective responsibility. In this sense, they are not believers but nor clearly hesitants. According to Kwok and colleagues [27], those HCWs are neutral in their vaccination attitudes. Considering our results, probably, they search for information about the vaccine, developing an average confidence towards the vaccine and sense of collective responsibility. The third profile was the hesitant group of HCWs and showed a configuration that is not in line with any of the previous studies. In fact, Leung and colleagues [23] identified a similar profile, the outsiders, with low calculation, confidence, and collective responsibility, and high constraints and complacency. Our profile was similar except for the calculation as in our sample we found above-average level of calculation for this profile. In this sense, those HCWs do not believe in the vaccination and are actively engaged in searching for information about the vaccine. According to Smith [43], individuals in this kind of profile are “refusers”, “late/selective vaccinators”. We may speculate that in HCWs this may be to some degree related the progressive emergence of SARS-CoV-2 variants of concern, to the (mis)concepts/knowledge about protection from infection vs. protection from severe disease and to fear for “waning immunity”. Finally, the fourth profile was the rejector group, and was in line with the skeptic profile previously described in the literature [23,27], showing the highest levels of calculation, complacency, and constraints. Our results showed that rejectors are high calculators, confirming Betsch and colleagues’ suggestion that “high calculation can lead to non-vaccination due to the high availability of anti-vaccination sources” [14]. Overall, the empirical identification of these profiles is in line with the SAGE definition of hesitancy that covers a broad spectrum of hesitant individuals [13]. Accordingly, vaccine hesitancy was defined as a continuum from acceptance to refusal of vaccines or as a delay in acceptance or refusal despite the availability of the vaccines [12,14].

When we inspected the relationship of these profiles with predictors, we found that having a history of vaccines was a significant predictor of believer and middler profiles when compared to the rejecter profile. These results were partially in line with previous studies, who found that believers were more likely to have been vaccinated against the seasonal flu than the other profiles [23,27]. Furthermore, concerning serological tests, our results suggested that it was associated to higher likelihood of being in the middler profile.

Concerning being exposed to having family\friends\colleagues that died of COVID-19, we found that the higher HCWs were exposed, the lower was the likelihood to be in the rejector profile. In this sense, direct experience of relatives died from COVID-19 may represent a crucial predictor of vaccine hesitancy. Although we are not aware of any study investigating this predictor of latent profiles, in a recent variable-center study [44] results suggested than those who had at least one relative/friend dying from COVID-19 were more likely to be hesitant. More studies are needed to confirm our results in a person-center perspective. Finally, showing high levels of conspiracy beliefs decreased the likelihood of being in the believer profile when compared to the other three profiles. In this sense, our results are in line with previous variable-centered studies [18,45] that highlighted how conspiracy about the COVID-19 was a significant predictor of vaccine hesitancy.

When we considered the likelihood of intention to get a fourth dose of COVID-19 vaccine in relationship with the profiles, we found that being in the believer profile has the highest probability to get vaccinated, whereas the lowest were in the hesitant and rejecter profiles. Our results are supported by other studies [23,27], suggesting that believers were shown to have the highest intention to have the booster dose when compared to the other profiles. In this sense, HCWs in this profile showed the highest probability to be confident in the safety and necessity of vaccines, and this confidence was associated to a stronger intention to vaccinate.

### 4.1. Implications

The results of our study can support the design and implementation of tailored intervention programs for hesitant subpopulations of HCWs. First, whereas high confidence and collective responsibility were crucial determinants of vaccination intention for the believer profile, for the middlers complacency, constraints, and conspiracy beliefs may be at the root of their reduced intention. Furthermore, the perceived reduced severity and mortality of the COVID-19 may have increased feelings of invulnerability. In this sense, as suggested by Leung and colleagues [23], implementing actions aimed at reducing conspiracy beliefs and increasing awareness about that vaccination is (still) necessary for HCWs for reduce the risk of severe disease, to further foster a higher vaccination rate in this profile. Concerning hesitant and rejecter profiles, they could be the hardest to convince to get vaccinated. In fact, according to Leung and colleagues [23], “they may have already formed a stable attitude about vaccines” that is hard to modify.

### 4.2. Limitations

Albeit our study contributes to profiling attitudes towards vaccination, we are aware of several methodological limitations. Firstly, the cross-sectional nature of the data does not allow us to draw some conclusions about the directionality of the associations between the observed profiles and the outcome variables. Longitudinal research is needed to determine whether profiles will emerge\change over time. Secondly, the consideration of potential antecedent and outcome variables was limited. In fact, by taking the variable-centered approach into account, many other antecedents (previous COVID-19 vaccination history and being infected) and outcomes (such as mandatory vaccination) would need further investigation in future research. Third, we relied on a convenience sample of Italian HCWs, which cannot be considered to be representative of the population of Italian HCWs. Thus, additional person-centered research should be performed to assess the generalizability of the identified profiles and of their associations with covariates. Finally, the 5C model was developed for investigating vaccine hesitancy in a pre-pandemic scenario. More research is needed for understanding if this model is suitable for Covid-19 vaccine booster(s) hesitancy.

## Figures and Tables

**Figure 1 vaccines-11-00273-f001:**
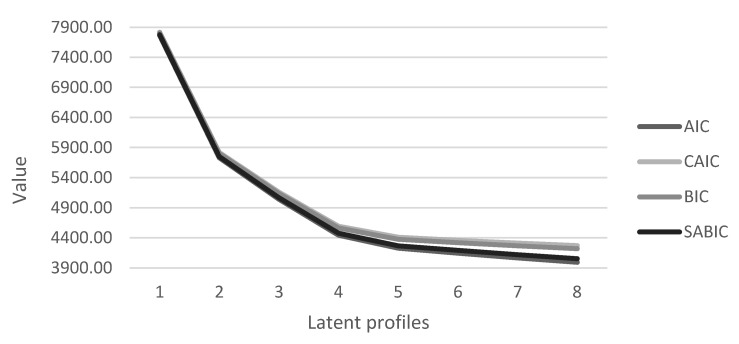
Elbow plot of the information criteria.

**Figure 2 vaccines-11-00273-f002:**
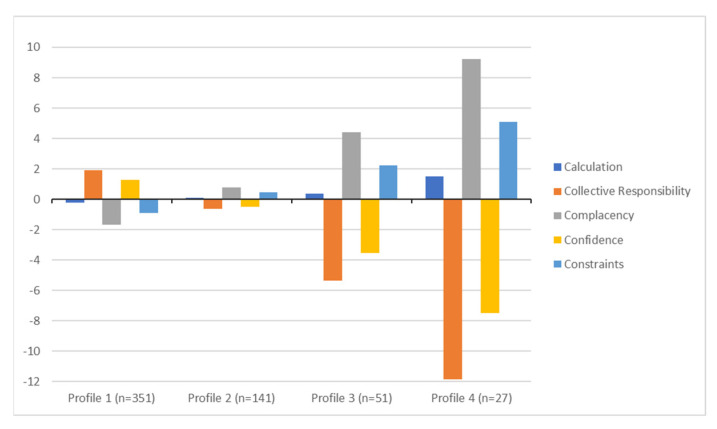
Results from the 4-profiles solution of 5C (values are estimated from factor scores with a mean of 0 and a standard deviation of 1).

**Table 1 vaccines-11-00273-t001:** Fit indices, entropy, and model comparisons for estimated latent profile analyses models.

Model	LL	#fp	Scaling	AIC	CAIC	BIC	SABIC	Entropy	aLMR	BLRT	AWE
1 Profile	−3871.94	10	1.739	7763.88	7817.33	7807.33	7775.59				11,721.00
2 Profiles	−2849.95	16	1.775	5731.89	5817.42	5801.42	5750.63	0.989	<0.001	<0.001	8718.13
3 Profiles	−2497.97	22	2.194	5039.93	5157.53	5135.53	5065.69	0.972	ns	<0.001	7725.30
4 Profiles	−2193.23	28	1.730	4442.46	4592.14	4564.14	4475.25	0.948	<0.001	<0.001	6874.21
5 Profiles	−2079.77	34	1.624	4227.55	4409.30	4375.30	4267.37	0.946	<0.001	<0.001	6596.95
6 Profiles	−2032.34	40	1.519	4144.68	4358.51	4318.51	4191.52	0.891	<0.001	<0.001	6517.76
7 Profiles	−1986.93	46	1.560	4065.87	4311.77	4265.77	4119.74	0.901	ns	<0.001	6444.65
8 Profiles	−1944.86	52	1.634	3993.73	4271.70	4219.70	4054.62	0.899	ns	<0.001	6381.55

Note: LL = log-likelihood; #fp = number of free parameters; AIC = Akaike Information Criterion; CAIC = Consistent Akaike Information Criterion; BIC = Bayesian Information Criterion; SABIC = Sample-size adjusted BIC; BLRT = bootstrapped likelihood ratio test; aLRT = Vuong–Lo–Mendell–Rubin adjusted likelihood ratio test; AWE = Approximate Weight of Evidence Criterion.

**Table 2 vaccines-11-00273-t002:** Predictors of latent profiles.

	1 vs. 4			2 vs. 4			3 vs. 4			2 vs. 1			3 vs. 1			3 vs. 2		
	Coef.	(S.E.)	OR	Coef.	(S.E.)	OR	Coef.	(S.E.)	OR	Coef.	(S.E.)	OR	Coef.	(S.E.)	OR	Coef.	(S.E.)	OR
FLU (No vs. Yes)	−1.17	1.26	0.31	−0.87	1.22	0.42	0.04	1.14	1.04	0.30	0.34	1.35	−1.21 *	0.61	3.36	0.91	0.53	2.48
HBV (No vs. Yes)	2.49 *	1.01	12.05	2.00 *	0.93	7.41	1.61	0.82	4.99	−0.49	0.42	0.62	−0.88	0.60	0.41	−0.40	0.47	0.67
SIER (No vs. Yes)	0.84	0.99	2.32	0.49	0.94	1.62	1.11	0.85	3.04	0.80 *	0.37	2.23	0.98	0.58	2.67	0.18	0.47	1.20
DEATH (No vs. Yes)	0.82	0.89	2.28	1.62 *	0.82	5.07	1.80 *	0.72	6.08	−0.36	0.34	0.70	0.27	0.58	1.31	0.63	0.50	1.87
VCBS	−6.90 **	0.75	0.00	−3.73 *	0.61	0.02	−2.14 *	0.54	0.12	3.17 *	0.42	23.81	4.76 *	0.53	116.96	1.59 **	0.29	4.91

Coefficients were estimated based on R3STEP logistic regression analyses. A positive (negative) coefficient value means that people with a higher score on this antecedent variable are more (less) likely to belong to the first latent profile among two profiles being compared. OR = odds ratios; S.E. = Standard error. * *p* < 0.05, ** *p* < 0.01 (two-tailed).

**Table 3 vaccines-11-00273-t003:** Probabilities of intention to get a fourth dose of COVID-19 vaccine across the four profiles.

		Probability (%)	S.E.
Profile 1: Believer (*n* = 351)			
Intention to get vaccine:	No	0.084	0.02
	Does not know	0.112	0.02
	Yes	0.804	0.03
Profile 2: Middler (*n* = 141)			
Intention to get vaccine:	No	0.279	0.04
	Does not know	0.237	0.04
	Yes	0.483	0.04
Profile 3: Hesitant (*n* = 51)			
Intention to get vaccine:	No	0.678	0.07
	Does not know	0.211	0.06
	Yes	0.111	0.04
Profile 4: Rejector (*n* = 27)			
Intention to get vaccine:	No	1	0
	Does not know	0	0
	Yes	0	0

Note: s.e.= standard error.

**Table 4 vaccines-11-00273-t004:** Equality tests of intention to get a fourth dose of COVID-19 vaccine probabilities across the 4 profiles.

	Chi-Square	*p*-Value	df
Overall test	3365.89	<0.001	6
Profile 1 vs. 2	34.67	<0.001	2
Profile 1 vs. 3	182.81	<0.001	2
Profile 1 vs. 4	2725.76	<0.001	2
Profile 2 vs. 3	41.63	<0.001	2
Profile 2 vs. 4	333.00	<0.001	2
Profile 3 vs. 4	24.42	<0.001	2

Note: df = degree of freedom.

## Data Availability

Data and materials are available from the corresponding author upon reasonable request.

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
