# Peer review of "COVID-19 Vaccine Hesitancy among Italian Healthcare Workers: Latent Profiles and Their Relationships to Predictors and Outcome"

_vaccines, 2023, doi:10.3390/vaccines11020273_

Round 1

Reviewer 1 Report

Dear authors, I read with interest your manuscript " COVID-19 vaccine hesitancy among healthcare workers: latent 2 profiles and their relationships to predictors and outcome

Here are some comments and suggestions 

Title: Since the study population is Italian HCW's please mention in the tile to that readers can have an idea which study population we are talking about. 

suggestion " COVID-19 vaccine hesitancy among Italian healthcare workers: latent 2 profiles and their relationships to predictors and outcome." 

Abstract:  Ok 

Introduction: well done. 

Material and Methods: 

110-112 : The authors used convenience snowball sampling on social media. You might add technical terms like convenience snowball sampling in the description here. 

Add the survey tool in the supplementary material 

Results: ok

Discussion : ok 

274 " To our knowledge, this is the first study to investigate latent profiles of hesitancy 274 towards booster dose of COVID-19 vaccine among western HCWs." correct it to Italian HCW's 

Limitation: ok 

Thanks 

Reviewer 2 Report

The language used in this manuscript is at times heavily weighted, judgemental, and biased. The introduction and discussion require the most extensive revisions to overcome this flaw in the manuscript.

line 39 - 'widely documented' is not supported with a single study. 

The introduction is descriptive and lacking criticality throughout.

Outdated models have been applied inappropriately to a new type of vaccine - all the research cited in the manuscript on vaccine hesitancy that was published pre-COVID is unlikely to apply to the novel mRNA vaccine that has yet to undergo the same stringent trials of more traditional vaccines. There is no recognition of this major factor in the limitations section of this report.

lines 73-74 - language such as 'conspiracy beliefs' is unjustified, undefined and judgemental labelling, as is language like middlemen (line 92).

The manuscript refers repeatedly to a 'person-centred approach' being adopted in this study. There is no evidence for this iuse of language anywhere in the document, that simply reports on statistical modelling and profiling. There is nothing within the report that remotely resembles a person-centred approach to this topic.

lines 267-270 - this set of outcomes is overstated, unremarkable, and unsuprising correlation, offering little in the way of advancing knowledge in this field of study.

line 327 - please define what is meant by 'high conspiracy beliefs'

Round 2

Reviewer 2 Report

The points raised in the first review have not been adequately addressed.

The discussion should include greater emphasis surrounding some of the methodological flaws associated with the study, including, for example, the use of instruments (e.g. vaccine conspiracy scale) that have not been adapted or validated for use in the context of the novel COVID-19 vaccines.

If the study insists on being referred to as adopting a 'person-centred' approach then it should adhere more closely to it by ensuring that the overall message of the paper advocates the individuals' needs, preferences, and values are listened to, heard, respected and responded to. This could be achieved by editing out (or at minimum showing a critical appreciation of) the value-laden and/or loaded labelling that is used in this (and cited) research with terms (such as 'conspiracy theorist' 'skeptic' and 'complacent') that all hold negative connotations. 

The level of critical insight in the discussion could be improved by offering alternative explanations/theory/evidence as to why HCWs might be vaccine 'hesitant' beyond those few factors included in the analysis. Extraneous variables (experiencing/hearing adverse events, health-related conditions (autoimmune factors), religious and spiritual beliefs, and so on) have not been considered.

The discussion could also offer awareness of the possibility that HCWs are potentially more attuned to some of the legitimate concerns around receiving an experimental mRNA injection that is yet to undergo long-term efficacy/safety trials. These points have not been raised in the discussion, but warrant consideration to help balance-out the overall message this paper puts across.

The discussion does not currently consider 'alternative' explanations for hesitancy beyond 'rejectors' being 'complacent conspiracy theorists lacking a collective responsibility'. 

Author Response

Reviewer’s points

Authors’ responses

The discussion should include greater emphasis surrounding some of the methodological flaws associated with the study, including, for example, the use of instruments (e.g. vaccine conspiracy scale) that have not been adapted or validated for use in the context of the novel COVID-19 vaccines.

We do not agree with this comment. The Vaccine Conspiracy Beliefs Scale is a well-developed instrument aimed at measuring general beliefs about vaccines. In this sense, it does not require any adaptation to the present context. Furthermore, in the COVID-19 scenario, it is a largely used instrument in general and occupational population. For example, here is a very brief list of national and wide studies where this measure was used:

1.     Freeman, D., Waite, F., Rosebrock, L., Petit, A., Causier, C., East, A., ... & Lambe, S. (2022). Coronavirus conspiracy beliefs, mistrust, and compliance with government guidelines in England. Psychological medicine52(2), 251-263.

2.     Freeman, D., Loe, B. S., Chadwick, A., Vaccari, C., Waite, F., Rosebrock, L., ... & Lambe, S. (2022). COVID-19 vaccine hesitancy in the UK: the Oxford coronavirus explanations, attitudes, and narratives survey (Oceans) II. Psychological medicine52(14), 3127-3141.

3.     Al-Sanafi, M., & Sallam, M. (2021). Psychological determinants of COVID-19 vaccine acceptance among healthcare workers in Kuwait: a cross-sectional study using the 5C and vaccine conspiracy beliefs scales. Vaccines9(7), 701.

4.     Ruiz, J. B., & Bell, R. A. (2021). Predictors of intention to vaccinate against COVID-19: Results of a nationwide survey. Vaccine39(7), 1080-1086.

If the study insists on being referred to as adopting a 'person-centred' approach then it should adhere more closely to it by ensuring that the overall message of the paper advocates the individuals' needs, preferences, and values are listened to, heard, respected and responded to. This could be achieved by editing out (or at minimum showing a critical appreciation of) the value-laden and/or loaded labelling that is used in this (and cited) research with terms (such as 'conspiracy theorist' 'skeptic' and 'complacent') that all hold negative connotations.

As we stated in the previous review process, the “person centered approach” refers to the statistics\methodological approach adopted in analyzing data. We do not understand what kind of person-centered approach the Anonymous reviewer refers to.

It is an internationally recognized method. According to Meyer, Stanley, and Vandenberg (2013), “The person-centered approach is so labeled because it takes into account intra-individual variation within a system of variables (Marsh, Lüdke, Trautwein, & Morin, 2009). That is, it acknowledges that variables can combine differently for some types of individuals than they do for others. Thus, rather than focusing on the variables per se, and how they relate within the population as a whole, person-centered research identifies and compares subgroups of individuals sharing similar patterns of variables within a population. Individuals are assigned to subgroups based on a configuration of variables and are therefore viewed from a more holistic perspective than is the case in variable-centered research (Vandenberg & Stanley, 2009).”

Concerning the (mis)use of some words, we did not judge anyone. Furthermore, in our manuscript there is no mention about “conspiracy theorists”. We presented scientific references about conspiracy theories.

Concerning the words “skeptic” and “complacent”, we just cited results from other studies where the word “skeptics” was used. We did not use this label, preferring a more neutral labelling.

Concerning the word “complacent”, we did not find it in our manuscript. Please, provide us evidence of it (lines).

The only word that is similar to it is “complacency”, the sub-dimension of the 5C measure. Again, it is an international measure, used and validated in many published studies. For instance, the following are just 3 recent studies,out of more that 20 using 5C, all taken from “Vaccines” journal:

Psychological Antecedents of Healthcare Workers towards Monkeypox Vaccination in Nigeria.

Ghazy RM, Okeh DU, Sallam M, Hussein M, Ismail HM, Yazbek S, Mahboob A, Abd ElHafeez S.

Vaccines (Basel). 2022 Dec 15;10(12):2151. doi: 10.3390/vaccines10122151.

Susceptibility towards Chickenpox, Measles and Rubella among Healthcare Workers at a Teaching Hospital in Rome. La Torre G, Marte M, Imeshtari V, Colaprico C, Ricci E, Shaholli D, Barletta VI, Serruto P, Gaeta A, Antonelli G.

Vaccines (Basel). 2022 Sep 20;10(10):1573. doi: 10.3390/vaccines10101573.

The Role of Psychological Factors and Vaccine Conspiracy Beliefs in Influenza Vaccine Hesitancy and Uptake among Jordanian Healthcare Workers during the COVID-19 Pandemic.

Sallam M, Ghazy RM, Al-Salahat K, Al-Mahzoum K, AlHadidi NM, Eid H, Kareem N, Al-Ajlouni E, Batarseh R, Ababneh NA, Sallam M, Alsanafi M, Umakanthan S, Al-Tammemi AB, Bakri FG, Harapan H, Mahafzah A, Al Awaidy ST.Vaccines (Basel). 2022 Aug 19;10(8):1355. doi: 10.3390/vaccines10081355.

The level of critical insight in the discussion could be improved by offering alternative explanations/theory/evidence as to why HCWs might be vaccine 'hesitant' beyond those few factors included in the analysis. Extraneous variables (experiencing/hearing adverse events, health-related conditions (autoimmune factors), religious and spiritual beliefs, and so on) have not been considered. The discussion could also offer awareness of the possibility that HCWs are potentially more attuned to some of the legitimate concerns around receiving an experimental mRNA injection that is yet to undergo long-term efficacy/safety trials. These points have not been raised in the discussion, but warrant consideration to help balance-out the overall message this paper puts across.

We totally disagree with this point. We can not speculate about the “extraneous” reasons behind HCWs hesitancy.

Factors like experiencing/hearing adverse events, health-related conditions (autoimmune factors), religious and spiritual beliefs, are linked to the intention to be vaccinated. We measured beliefs related to vaccination.

Furthermore, the scientific literature suggested that “Factors explaining suboptimal vaccination attitudes among HCWs [6–9] include misinformation, loss of confidence, fear of adverse effects, absence of educational campaigns, inaccurate risk perception, unknown or uncertain vaccination status and difficulties in accessing vaccination in the workplace.” (Bianchi, Stefanizzi, Brescia, Lattanzio, Martinelli, & Tafuri, 2022). 

Concerning “legitimate concerns around receiving an experimental mRNA injection that is yet to undergo long-term efficacy/safety trials”, this comment is not suitable for our study as we considered only intention to get booster vaccination. Furthermore, we did not ask for vaccination status, as the vaccination among the Italian population was close to 90% and mandatory vaccination (first cycle) 97% among HCWs (Task force COVID-19 del Dipartimento Malattie Infettive e Servizio di Informatica, Istituto Superiore di Sanità. Epidemia COVID-19. Aggiornamento nazionale: 14 dicembre 2022). Raising points about efficacy\safety of the covid vaccination is far from the study objectives.
